# Disparities in Smokefree and Vapefree Home Rules and Smokefree Policy Attitudes Based on Housing Type and Cigarette Smoking Status, United States, 2019

**DOI:** 10.3390/ijerph20146356

**Published:** 2023-07-13

**Authors:** Carolyn M. Reyes-Guzman, Minal Patel, Teresa W. Wang, Nalini Corcy, Dana Chomenko, Beth Slotman, Robert E. Vollinger

**Affiliations:** 1Tobacco Control Research Branch, Division of Cancer Control and Population Sciences, National Cancer Institute, Bethesda, MD 20892, USA; 2American Cancer Society, Atlanta, GA 30303, USA; 3Office on Smoking and Health, National Center for Chronic Disease Prevention and Health Promotion, Centers for Disease Control and Prevention, Atlanta, GA 30333, USA; 4BLH Technologies, Inc., Rockville, MD 20850, USA; 5Westat, Rockville, MD 20850, USA

**Keywords:** smoking, multi-unit housing, residents, secondhand or environmental exposure, electronic cigarettes, vaping, tobacco policies

## Abstract

This study examined variations in cigarette smoking status, home smoking and vaping rules, and attitudes toward smoking rules among U.S. adults. We analyzed data from the 2019 U.S. Census Bureau’s Current Population Survey Supplements (*n* = 40,296 adults) and calculated weighted prevalence estimates of adult cigarette smoking based on housing type. In 2019, multi-unit housing (MUH) residents who currently smoked were predominantly residents of privately rented housing (66.9%), followed by privately owned (17.6%) and public housing (15.5%). MUH residents who currently smoked had the highest proportions of allowing smoking (26.7%) or vaping (29.1%) anywhere inside their homes and were least likely to support rules allowing smoking inside all MUH apartments or living areas. In the adjusted models, MUH residents with a current smoking status were 92% less likely to have a complete smoking ban. More than one in four MUH residents with a current smoking status allowed all smoking inside the home and supported allowing smoking inside all MUH apartment or living areas, reinforcing how MUH residents may be at higher risk of experiencing secondhand smoke or aerosol exposure, or incursions within their places of residence. Our results can inform the development, implementation, and sustainment of strategies to reduce exposures from tobacco and nicotine products in all living environments.

## 1. Introduction

The harms associated with secondhand smoke (SHS) exposure, including increased risk of respiratory illness, cancer, and cardiovascular disease, are well documented [1]. Smokefree air laws help reduce exposure to SHS [2,3], and they can be an effective tool in reducing smoking prevalence [4,5], resulting in improved health outcomes and reduced mortality for smoking-related illnesses [2,6]. Smokefree policies have also been shown to decrease the social acceptance of smoking, leading to increased quitting intentions [7], and to promote the adoption of voluntary home smoking bans and smokefree environments [8]. These adopted home rules have proven a powerful tool for successful cessation among adults [9] and for further preventing smoking initiation by youth [10]. However, there is inequitable access to and adoption of voluntary smokefree policies among residents in single-unit housing (SUH), defined as independent housing units that do not share a wall or ceiling, compared with multi-unit housing (MUH), defined as two or more housing units that share a wall or ceiling. These inequities are associated with disproportionate tobacco-related health disparities [11]. Characterizing adults’ behaviors and attitudes around commercial tobacco—excluding traditional or ceremonial tobacco—such as home smoking or vaping rules and attitudes on indoor smokefree policies, may help accelerate public health progress in eliminating tobacco use-related death and disease. Notably, compared to SUH residents, individuals residing in MUH may be at a disproportionately increased burden for SHS exposure or secondhand aerosol (e.g., through electronic nicotine delivery systems (ENDS), also referred to as e-cigarettes or vaping) emissions from tobacco products due to infiltration from nearby units, even when they maintain smoking bans inside their homes [12,13,14,15,16]. Some studies particularly highlight differences in SHS exposure among MUH residents, suggesting that younger adults and historically underserved groups, such as Black and Hispanic individuals and individuals living below the Federal poverty level, are at a higher risk of being involuntarily exposed to SHS in their homes [17,18,19,20,21,22,23,24]. For example, 67.9% of Black children were found to be exposed to SHS at home compared to 37.2% of White children [23], and exposure to SHS increases as socioeconomic status decreases [25].

While many state and local regulations have restricted indoor exposure to SHS from combustible tobacco products—but not aerosol from electronic ENDS—in workplaces, restaurants, and bars, the only places that currently have enacted ordinances to limit smoking in MUH (as of 1 July 2019) are cities or counties in California [26]. Federally, the U.S. Department of Housing and Urban Development (HUD) implemented a rule in 2017 requiring all federal public housing properties to be 100% smokefree indoors and within 25 feet of buildings [27]. Currently, approximately 26% of all U.S. residents, including adults and children, reside in publicly owned (or subsidized) and privately owned (or market share) MUH [28], so examining smoking in MUH beyond the HUD rule is important, as HUD’s policy only applies to smoking in multi-unit public housing [29,30,31]. The HUD rule therefore only covers a quarter of all MUH residents and does not apply to the vast majority of residents living in private MUH who are likely to be from historically disadvantaged populations and who may be exposed to SHS from neighbors, even if their own household is smokefree or vapefree.

As states and local governments consider implementing similar smokefree policies, it is important to understand smoking prevalence and other factors related to MUH compared to SUH. Based on the conceptual model by Swope and Hernandez, housing conditions can play an important role in health equity [32]. The current study serves to address research gaps from other studies which were largely limited to specific subpopulations, geographies, and/or housing types. To do so in a comprehensive manner, this study examines cigarette smoking status, smokefree and vapefree home rules, and attitudes towards smokefree policies at the national level based on sociodemographic and housing characteristics.

## 2. Methods

### 2.1. Data Source

We linked and analyzed two cross-sectional supplements of the household-based Current Population Survey (CPS), which use probability-based, multi-stage sampling; the March 2019 Annual Social and Economic Supplement (ASEC); and the January and May 2019 Tobacco Use Supplement to the CPS (TUS-CPS) [33]. These supplements include a nationally representative study population of non-institutionalized U.S. adults aged ≥18 years who completed the surveys once either by phone (two-thirds of participants) or at home (one-third of participants). In total, the final merged analytic sample included 40,956 participants, further reduced to 40,296 due to missing observations on smoking and housing type variables (self-response rate was 58.9% for January 2019 TUS, 56.2% for May 2019 TUS, and 67.6% for 2019 ASEC). The findings reflect smoking-related behaviors among an estimated 59.9 million adult MUH residents and 162.3 million SUH residents in the U.S. population.

### 2.2. Measures

#### 2.2.1. Smoking Status

Smoking status was defined by smoking at least 100 cigarettes in the lifetime and currently smoking every day or some days (those who currently smoked), not smoking at all in the last 30 days (those who formerly smoked), and not meeting the lifetime 100 cigarette threshold (those who never smoked).

#### 2.2.2. Housing Type

Housing type was dichotomized as MUH, defined as ≥2 joined housing units such as apartments, duplexes, and other housing units that share a wall or ceiling, versus SUH, defined as independent housing units such as single-family homes, mobile homes, and other housing units that do not share a wall or ceiling.

#### 2.2.3. Housing Tenure

Consistent with HUD terminology [29], housing tenure (i.e., housing ownership or tenancy) included three categories: (1) public, (2) privately owned or bought, and (3) privately rented, including no-cash rent or rent without payment.

#### 2.2.4. Smoking and Vaping Home Rules

Adults were asked about their home smoking rule using the question “Which statement best describes the rules about smoking inside your home?” (Response options were as follows: “No one is allowed to smoke anywhere”; “Smoking is allowed in some places or at some times”; and “Smoking is permitted anywhere”). A new question in this TUS-CPS cycle asked respondents about their ENDS use rule using the question “Which statement best describes the rules about vaping or using e-cigarettes inside your home?” (Response options were as follows: “No one is allowed to vape anywhere”; “Vaping is allowed in some places or at some times”; and “Vaping is permitted anywhere”).

#### 2.2.5. Smokefree MUH Policy Attitudes

All participants were also asked about their degree of support for an indoor MUH smoking ban inside apartments and common living areas (e.g., smoking should (1) be allowed inside all apartments or living areas, (2) be allowed inside some apartments, or (3) not be allowed at all inside apartments).

#### 2.2.6. Sociodemographic Variables

Sociodemographic variables included age (18–24, 25–34, 35–44, 45–54, and 55+ years old), race and ethnicity (White non-Hispanic; Black non-Hispanic; Other non-Hispanic, including American Indian and Alaska Native, Asian, Hawaiian/Pacific Islander, and two or more races; and Hispanic), sex (male, female), education (less than high school; high school/GED equivalent or less; some college, no degree; associate degree; and college degree or above), U.S. Census region (Northeast, Midwest, South, and West), annual household income (<USD 20,000, USD 20,000–USD 39,999, USD 40,000–USD 74,999, and USD 75,000+), metropolitan status (metropolitan/non-metropolitan, based on Census FIPS Metropolitan Area Core Based Statistical Area (CBSA) codes and collapsed for non-identifiability), employment status (employed, unemployed, and not in labor force), occupational category (white collar, blue collar, service, and other/not in labor force).

### 2.3. Statistical Analysis

We assessed the unadjusted weighted proportions of U.S. adults for (1) differences between sociodemographic and housing tenure variables, based on housing type and smoking status; (2) distributions of those who reported being covered by home smoking rules and home vaping rules; and (3) support for indoor MUH smoking bans, according to housing type, tenure, and smoking status, regardless of whether they lived in MUH or SUH. We calculated Chi-squared tests to compare differences in tobacco-related behaviors (i.e., home rules and attitudes for support) based on housing tenure, housing status, and smoking status (*p* < 0.05) [34].

We also fitted two weighted polytomous logistic regression models to examine the association between smoking status, and presence of home smoking and vaping rules, separately. Response categories for each home rule variable were grouped as having a complete ban versus a partial/no ban on either smoking or vaping inside the home, separately among residents of MUH and SUH. Categories for “partial ban” and “none at all” were grouped as our main interest was in examining individuals reporting a complete ban. Covariates in both models included the aforementioned sociodemographic, labor, and housing tenure variables. We excluded the occupational category variable from final models due to collinearity with employment status.

All analyses used weights from the TUS-CPS (not the ASEC supplement) as the unweighted and weighted self-response counts for the overlapping TUS-ASEC samples were 91.4% and 90.8%, respectively (with a similar overlap based on age, gender, race, and ethnicity), indicative of a large overlap of respondents from the two CPS supplement samples. This overlap supported retaining self-response weights from only TUS-CPS instead of recalculating a new weight based on both TUS and ASEC weights. We conducted analyses using SAS-callable SUDAAN (SAS version 9.4 (SAS Institute); SUDAAN version 11.0.3 (RTI International)), applying self-response survey weights and replicate weights from the TUS-CPS.

## 3. Results

Overall, in 2019, 24% of U.S. adults lived in MUH and 76% lived in SUH. About 11% of U.S. adults living in either MUH or SUH currently smoked cigarettes, while among MUH residents, 15.6% formerly smoked (95% CI, 14.1–17.2) and 73.6% never smoked (95% CI, 71.7–75.5) cigarettes. Among SUH residents, 19.3% formerly smoked (95% CI, 18.4–20.3) and 70.0% never smoked (95% CI, 68.7–71.2) (Table 1). About one-quarter (27.2%) of adults who currently smoked cigarettes were MUH residents.

More than half of all adults who currently smoked and lived in MUH were men (55.0%), 36.2% were aged 55 years or older; 54.2% were White non-Hispanics; 31.3% lived in the South; 91.3% resided in metropolitan areas; 34.0% reported an annual household income of less than USD 20,000; 38.6% possessed a high school education; 55.5% were employed; 29.6% reported a white collar occupational category; and 66.9% lived in privately rented housing (Table 1 and Appendix A). Among MUH residents, 17.6% of those who currently smoked, 33.2% who formerly smoked, and 25.0% who never smoked lived in privately owned housing, while there was a larger proportion of adults who currently smoked living in public housing compared to those formerly or never smoked (15.5%, 9.5%, and 7.5%, respectively). The patterns of sociodemographic characteristics among adults who currently smoked and lived in SUH were similar to MUH residents, except when examining income and housing tenure. The majority of adults who currently smoked and lived in SUH reported higher annual household income ranges of USD 40,000–USD 74,999 (30.1%) and USD 75,000 or higher (26.7%) and reported living in privately owned homes (69.2%).

### 3.1. Presence of and Support for Smokefree Home Rules

Figure 1 presents the distributions of respondents who reported whether they allowed cigarette smoking inside their homes (i.e., home smoking rules) and whether they supported indoor smoking bans in MUH, regardless of whether they themselves lived in MUH or SUH. These comparisons are stratified by housing tenure and smoking status. Among MUH residents, 10.8% of adults living in public housing reported allowing smoking in all areas of their homes, versus 5.2% living in privately rented housing and 3.8% living in privately owned housing (Figure 1, Appendix A). Similar patterns were observed among SUH residents (10.2%, 6.1%, and 4.1%, respectively). Further, 26.7% of MUH residents who currently smoked allowed smoking in all areas of their homes, followed by 4.8% who formerly smoked, and 2.4% who never smoked; we observed a similar pattern, but with slightly lower proportions, among SUH residents (23.2%, 3.8%, and 1.9%, respectively). With respect to support for indoor smoking bans, approximately 70% of all MUH residents reported support for “not at all allowing indoor smoking in [MUH] buildings”, regardless of whether they lived in public, privately owned, or privately rented housing. In contrast, we found significant differences in support for indoor bans by smoking status; moderate proportions of MUH residents who currently smoked supported allowing all smoking (28.2%) or some smoking (30.1%) indoors. In contrast, 67.2% of MUH residents who formerly smoked and 73.9% of those who never smoked supported not allowing smoking indoors. We observed similar patterns among SUH residents, with 73.3% of those who never smoked expressing support versus 61.6% of those who formerly smoked and 45.9% of those who currently smoked.

### 3.2. Vapefree Home Rules

Figure 2 focuses on adults who reported whether they allowed vaping inside their homes. More than 84% of adults resided in MUH or SUH homes that did not allow ENDS use at home (Figure 2, Appendix A). Among MUH residents, 11.0% of adults living in public housing allowed vaping in any area inside their homes versus 7.8% living in privately rented housing and 5.1% in privately owned housing, similar to SUH residents (10.1%, 7.9% and 5.1%, respectively). Among MUH residents, 29.1% of those who currently smoked allowed vaping in any area inside their homes, followed by 9.7% of those who formerly smoked and 3.8% of those who never smoked, compared to 26.1%, 6.6%, and 2.4%, respectively, among SUH residents.

### 3.3. Multivariable Modeling

Table 2 summarizes the results for two multinomial logistic regression models, presented as three sets of paired comparisons to the reference group, for each model. The reference groups were “SUH with a partial or no smokefree rule” and “SUH with a partial or no vapefree rule” at home, respectively (reference group OR = 1 for each model not shown; full results in Appendix A). The results are described according to each paired comparison.

#### 3.3.1. Determinants of MUH Complete Smokefree Rules

Among MUH residents, Hispanic adults were 2.46 times significantly more likely (95% CI, 1.40–4.32) to report a complete smokefree home rule (i.e., no smoking allowed inside the home) versus White non-Hispanics. Among MUH residents, complete smokefree rules were also more likely to be reported among those living in the Northeast and West U.S. regions compared with the Midwest (OR, 2.12; 95% CI, 1.35–3.32 and OR, 1.86; 95% CI, 1.19–2.90, respectively), metropolitan areas (OR, 3.41; 95% CI, 2.24–5.17) compared with those in non-metropolitan areas, and publicly rented (OR, 63.25; 95% CI, 18.48–216.48) or privately rented MUH (OR, 8.35; 95% CI, 5.83–11.94) compared with privately owned MUH (Table 2a). Compared to individuals living in MUH with a college degree, those with less than a high school education had 75% lower odds of having a complete smokefree housing rule (OR, 0.25; 95% CI, 0.15–0.41).

#### 3.3.2. Determinants of MUH Partial or No Smokefree Rules

The odds of reporting partial or no smokefree home rules were higher among those living in the Northeast U.S. region (OR, 2.50; 95% CI, 1.37–4.55) compared to adults living in the Midwest. MUH residents with a partial or no rule showed similar results (of a lower magnitude) to MUH residents with a complete smokefree home rule, for those living in metropolitan areas, for those living in publicly or privately rented MUH, and based on educational status.

#### 3.3.3. Determinants of SUH Complete Smokefree Rules

Among SUH residents, Hispanic adults were 1.74 times more likely to report a complete smokefree home rule (OR, 1.74; 95% CI, 1.03–2.93) compared to White non-Hispanic adults. Odds were also higher among those living in the West (OR, 1.78; 95% CI, 1.23–2.59) compared to the Midwest. In contrast, odds of reporting complete smokefree rules in SUH were lower among those with household incomes lower than USD 75,000 (range: OR, 0.56; 95% CI, 0.38–0.84 for <USD 20,000 to OR, 0.73; 95% CI, 0.54–0.99 for USD 40,000–USD 74,999), compared to those with incomes USD 75,000 or higher. Similarly, odds of reporting complete smokefree rules were generally lower among those with the least educational attainment (range: OR, 0.51; 95% CI, 0.35–0.74 for those with less than high school education to OR, 0.71; 95% CI, 0.51–0.99 for those with some college, no degree) Similar to adults living in MUH with a complete rule, odds of living in SUH with complete smokefree rules were lower among those who currently smoked (OR, 0.09; 95% CI, 0.07–0.12) and formerly smoked (OR, 0.70; 95% CI, 0.53–0.91) compared to those who never smoked.

#### 3.3.4. Determinants of MUH Complete Vapefree Rules

Among MUH residents, odds of reporting complete vapefree home rules were lower among adults aged 18–24 years (OR, 0.44; 95% CI, 0.27–0.74), 25–34 years (OR, 0.56; 95% CI, 0.39–0.80), and 45–54 years (OR, 0.68; 95% CI, 0.48–0.96) compared to adults aged 55 or older; higher among Hispanics (OR, 2.63; 95% CI, 1.53–4.53) or Other non-Hispanics (OR, 1.76; 95% CI, 1.02–3.02) compared to White non-Hispanics; and higher in the Northeast (OR, 2.06; 95% CI, 1.31–3.25) compared to the Midwest (Table 2b). We observed similar patterns to MUH complete smokefree rules for metropolitan status, educational attainment, housing tenure, and smoking status.

#### 3.3.5. Determinants of MUH Partial or No Vapefree Rule

We observed similar patterns among MUH adults reporting partial/no home vapefree rules to describe patterns among MUH residents with complete vapefree home rules, for U.S. region, metropolitan status, educational attainment, and housing tenure.

#### 3.3.6. Determinants of SUH Complete Vapefree Rules

Similar to patterns among adults with a MUH complete vapefree rule (with a smaller magnitude of effect), the likelihood of living in SUH with a complete home vapefree rule was lowest among the youngest adults aged 18–24 (OR, 0.50; 95% CI, 0.31–0.79) compared to adults aged 55 or over, while Hispanic individuals were 1.83 times more likely (95% CI, 1.12–3.00) than White non-Hispanic individuals to report a complete vapefree rule. Additionally, adults with income <USD 75,000 and those who currently smoked or formerly smoked were less likely to report a complete vapefree rule at home compared to those with income USD 75,000 or higher and those who never smoked, respectively.

## 4. Discussion

This study leverages housing characteristics—a key determinant of health equity [32]—to pinpoint national-level differences in tobacco product-use-related behaviors. In 2019, adult MUH residents who currently smoked cigarettes were least likely to have complete smokefree or vapefree home rules when compared to MUH residents who formerly or never smoked. Furthermore, MUH residents who currently smoked were least likely to support no smoking inside buildings with multiple apartments or living areas, compared to those who formerly or never smoked. Our study also showed that, although the majority of MUH residents who currently smoked lived in privately rented (66.9%) or privately owned (17.6%) housing, a substantial proportion (15.5%) lived in publicly sponsored housing. Additionally, the majority of residents living in privately owned MUH never smoked (33.2%) or formerly smoked (25.0%), whereas 17.6% currently smoked. Multivariable modeling also showed that adults from the lowest educational groups, the Midwest U.S. region, non-metropolitan residents, who currently smoked and who privately owned or privately rented MUH were least likely to support a complete smokefree home rule.

The differences in support of indoor smoking bans by smoking status highlight a need for targeted smokefree policies and cessation interventions, especially in light of differences among populations experiencing health disparities. Smoking among MUH residents, especially given the higher (unadjusted) smoking prevalence among those in publicly owned dwellings, likely puts MUH residents at a higher risk for involuntary SHS and secondhand aerosol (SHA) exposure. However, our adjusted results also indicate that those living in publicly rented MUH were more likely to report a complete smokefree rule, which may be reflective of the HUD rule and its impact to effectively lower SHS exposure. Although the HUD rule only applies to a subset of MUH residents who live in public housing, our positive finding supports the adoption of policies in private MUH.

Notably, this study extends beyond the presence of smokefree home rules to examine the presence of vapefree home rules by housing type and housing tenure, which to our knowledge has not been explored yet includes key determinants of health equity [32]. These multivariable findings were generally similar to those reporting smokefree rules; however, we found that the younger age groups (18–34 years) were least likely to indicate presence of complete vapefree home rules, among both MUH and SUH residents. It is possible that these findings indicate a wider acceptability of vaping compared to cigarette smoking as a social norm. Research suggests that ENDS use may expose others to secondhand aerosol, which can contain harmful and potentially harmful constituents, including nicotine, heavy metals, ultrafine particulates, volatile organic compounds, and other toxicants [35]. Compared to SHS, far less is known about the health effects of secondhand ENDS aerosol given the relative recency of e-cigarettes on the market, but emerging research indicates that indoor ENDS use may expose non-users to increased concentrations of particulate matter and volatile organic compounds, though potentially at lower concentrations than cigarette smoke [1,35,36,37,38,39]. These results underscore the importance of efforts to promote smoking cessation and comprehensive smokefree and vapefree policy adoption in MUH.

Data from the National Adult Tobacco Survey in 2013–2014 indicated that tobacco product use was higher among individuals living in MUH (24.7%) than single-family housing (18.9%), and smokefree home rules were lower among MUH residents (80.9%) compared to SUH residents (86.7%) [17]. Though the findings from our study are not directly comparable due in part to differences in survey fielding, it is a more recent investigation indicating a generally lower prevalence of smoking at 11% for both MUH and SUH residents, which follows national trends in cigarette smoking reduction. Additionally, the novelty of our analysis lies in the further granularity of characteristics like housing type and tenure available in the ASEC supplement.

Interestingly, several recent studies have found strong support for smokefree policies among residents of MUH [40,41,42]. McMillen and colleagues found that a majority of Section 8 public housing residents (71%) supported prohibiting smoking everywhere in MUH, versus just in units with Section 8 subsidies (38%) [40] (Section 8 is a specific type of housing voucher, sponsored by the US Federal Government’s Department of Housing and Urban Development. See details here: https://www.hud.gov/topics/housing_choice_voucher_program_section_8 (accessed on 2 September 2022)). A more recent study by Patel and colleagues revealed that 77% and 74% of U.S. adults, including similar percentages of MUH residents, supported prohibiting cigarette use and e-cigarette use, respectively, in MUH, while approximately half of adults who currently smoked supported both policies [41]. From a health equity standpoint, several articles note that Hispanic individuals may be at a particularly disproportionate risk of SHS since a large proportion of Hispanic individuals reside in MUH [43,44]. Despite the presence of such disparities, our data show that both Hispanic MUH and SUH residents were more likely to report the presence of complete smokefree and vapefree home rules compared to White individuals.

As highlighted by the U.S. Surgeon General, private settings, including the home environment, remain a major source of SHS exposure, particularly for youth [44]. In addition to implementing smokefree air policies, facilitating equitable access to evidence-based cessation resources is important for maximizing the success of these policies. Successful smokefree air policies have been coupled with promotion of free cessation programs for individuals living in MUH; this is especially important given that access to cessation programs may be limited for historically underserved populations living in MUH [45]. In addition to progress at the federal level, as of 15 August 2020, more than 63 cities and counties in California have local laws requiring smokefree policies in all MUH, including both government or subsidized, and private market-rate housing [27]. Data have shown that the adoption and maintenance of household smokefree rules in private single-family homes and smokefree policies in subsidized and public MUH are associated with decreased consumption of cigarettes, increased confidence in achieving cessation, increased potentially considerable cost savings, and greater prevalence of successful cessation [9,46,47,48,49].

This study’s strength also includes the use of a large and nationally representative sample, leveraging two CPS supplements, which together provide detailed data on smoking status, and housing type and tenure. However, this study is subject to limitations. First, this study only examines cigarette smoking status and does not examine other tobacco product use, such as ENDS, based on housing type and tenure because TUS CPS did not include questions on attitudes towards ENDS use in MUH. Given that ENDS use continues to be high among youth and young adults [50,51,52,53], it will be important for future research to understand the prevalence of (as well as support for) comprehensive smokefree air policies that also include ENDS. Second, this study did not assess trends in smokefree air or vapefree home rules over time; this will be an important topic for future research. Third, this study does not assess household smoking, and therefore, does not account for interpersonal influences on smoking in the household. Fourth, our results cannot disentangle differences among respondents categorized as Other non-Hispanic due to small sample sizes in those subgroups, especially among persons reporting more than one racial group. Fifth, when examining the stratified data for smokers, we had small cell sizes, leading to larger confidence intervals, and less specific estimates. Sixth, this study could not differentiate between types of multi-unit housing; it is likely, for example, that there are differing characteristics of those living in duplexes compared to larger apartment buildings; however, examining MUH as a group is still important as the SHS from those not in their household can still impact their health. Finally, it should be noted that the income cutoff was USD 75,000 and there may be differences among people with incomes below the cutoff that are masked.

## 5. Conclusions

Smoking among residents of MUH, especially given the higher prevalence among those in publicly owned dwellings, puts non-smoking residents of MUH at a higher risk than SUH residents for involuntary SHS exposure, which in turn places MUH residents at higher risk for SHS-related morbidity and mortality. Residents living in private MUH are not covered by the HUD smoking rule, so opportunities exist to protect more individuals from SHS by extending smokefree policies to all MUH, including that which is privately owned or rented. Further, to promote health equity and to directly reduce tobacco product use among disparate populations that include racial and ethnic minorities and those with lower socioeconomic status, promoting cessation and ensuring equitable access to cessation services when implementing other commercial tobacco control interventions (such as smokefree policies) continue to be critical among individuals living in MUH, regardless of housing tenure. The results of this study can inform the development, implementation, and sustainability of effective, evidence-based interventions including increased access to evidence-based smoking cessation resources, as well as educational campaigns to lower social acceptance of SHS and SHA exposure and to highlight the benefits of smokefree and vapefree rules in shared spaces, irrespective of residence type.

## Figures and Tables

**Figure 1 ijerph-20-06356-f001:**
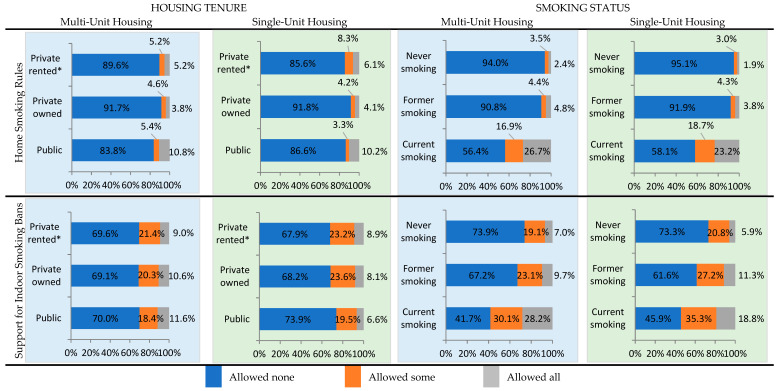
Home smoking rules and support for indoor multi-unit housing smoking bans among multi-unit housing and single-unit housing residents based on housing tenure and smoking status, 2019. Notes: 1. Percentages are weighted. 2. *p*-values for all chi-squared tests were statistically significant at *p* < 0.0001 (see Appendix A for details). 3. Number of respondents (unweighted) based on housing tenure—multi-unit housing: public, n = 873; privately owned, n = 2520; privately rented, n = 5865; single-unit housing: public, n = 185; privately owned, n = 24,720; privately rented, n = 4567. 4. Number of respondents (unweighted) by smoking status—multi-unit housing: Never smoked, n = 6421; Formerly smoked, n = 1756; Currently smoking, n = 1069; single-unit housing: Never smoked, n = 19,746; Formerly smoked; n = 6421. Currently smoking, n = 3252. 5. Multi-unit housing is defined as two or more housing units that share a wall or ceiling. Single-unit housing refers to independent housing units such as single-family homes, mobile homes, and other housing units that do not share a wall or ceiling. 6. Home smoking rules determined by analyzing responses to the following TUS-CPS question: Which statement best describes the rules about smoking inside your home? (FR: Read if necessary; “Home” is where you live. “Rules” include any unwritten “Rules” and pertain to all people whether or not they reside in the home or are visitors, workmen, etc. “Smoking” includes cigars, regular and hookah pipes, as well as cigarettes). (1) No one is allowed to smoke anywhere inside your home. (2) Smoking is allowed in some places or at some times inside your home. (3) Smoking is permitted anywhere inside your home. 7. Support for indoor smoking bans determined by analyzing responses to the following TUS-CPS question: In buildings with multiple apartments or living areas, do you THINK that smoking should be (1) Allowed inside all apartments or living areas, (2) Allowed inside some apartments, (3) Not allowed at all inside apartments? * Rent includes no cash rent or rent without payment.

**Figure 2 ijerph-20-06356-f002:**
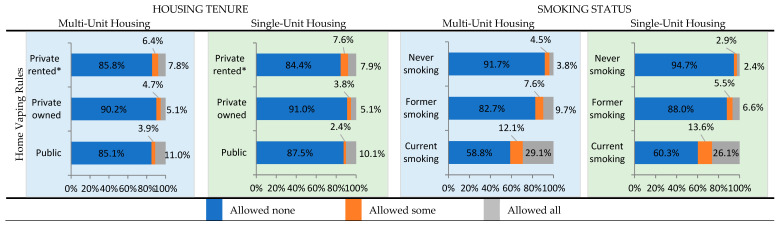
Home vaping rules among multi-unit housing and single-unit housing residents based on housing tenure and smoking status, 2019. Notes: 1. Percentages are weighted. 2. *p*-values for all chi-squared tests were statistically significant at *p* < 0.0001 (see Appendix A for details). 3. Number of respondents (unweighted) based on housing tenure—multi-unit housing: public, n = 873; privately owned, n = 2520; privately rented, n = 5865; single-unit housing: public, n = 185; privately owned, n = 24,720; privately rented, n = 4567. 4. Number of respondents (unweighted) based on smoking status—multi-unit housing: Never smoked, n = 6421; Formerly smoked, n = 1756; Currently smoking, n = 1069; single-unit housing: Never smoked, n = 19,746; Formerly smoked; n = 6421. Currently smoking, n = 3252. 5. Multi-unit housing is defined as two or more housing units that share a wall or ceiling. Single-unit housing refers to independent housing units such as single-family homes, mobile homes, and other housing units that do not share a wall or ceiling. 6. Home vaping rules determined by analyzing responses to the following TUS-CPS question: Which statement best describes the rules about vaping or using e-cigarettes inside your home? (FR: Read if necessary; “Home” is where you live. “Rules” include any unwritten “Rules” and pertain to all people whether or not they reside in the home or are visitors, workmen, etc. “Vaping” includes use of e-cigarettes, vape-pens, hookah-pens, ehookahs, e-vaporizers, vapes, or mods. popular brands in-clude njoy, blu, logic, vuse, and juul (prounounced like “Jewel”)). (1) No one is allowed to vape anywhere inside your home. (2) Vaping is allowed in some places or at some times inside your home. (3) Vaping is permitted anywhere inside your home. * Rent includes no cash rent or rent without payment.

**Table 1 ijerph-20-06356-t001:** Participant sociodemographic, occupation, and housing tenure characteristics among multi-unit and single-unit housing residents based on smoking status, 2019.

	Multi-Unit Housing				Single-Unit Housing			
Variable	Total N = 59,910,550 (n = 9662; 24%)	Current Smoking (n = 1151; 10.8%)	Former Smoking (n = 1850; 15.6%)	Never Smoking (n = 6661; 73.6%)	Total N = 162,349,276 (n = 30,634; 76%)	Current Smoking (n = 3484; 10.7%)	Former Smoking (n = 6797; 19.3%)	Never Smoking (n = 20,353; 70.0%)
	n	n (%)	n (%)	n (%)	n	n (%)	n (%)	n (%)
**Sex**								
Female	5374	584 (45.0)	901 (45.6)	3889 (55.4)	16,495	1712 (45.6)	3058 (43.3)	11,725 (54.7)
Male	4288	567 (55.0)	949 (55.4)	2772 (44.6)	14,139	1772 (54.4)	3739 (56.7)	8628 (45.3)
**Age Group**								
18–24	822	63 (9.3)	46 (4.1)	713 (18.3)	1158	98 (5.0)	60 (1.9)	1000 (12.2)
25–34	2196	203 (19.1)	227 (15.0)	1766 (27.8)	3924	501 (17.8)	499 (8.7)	2924 (16.0)
35–44	1536	198 (17.1)	250 (15.5)	1088 (15.6)	4972	639 (18.4)	835 (13.3)	3498 (17.0)
45–54	1281	197 (18.3)	203 (12.1)	881 (13.1)	4827	669 (20.3)	836 (15.4)	3322 (17.6)
55+	3827	490 (36.2)	1124 (53.4)	2213 (25.2)	15,753	1577 (38.6)	4567 (60.6)	9609 (37.3)
**Race/Ethnicity**								
Hispanic	1574	106 (13.6)	192 (15.4)	1276 (25.6)	2748	188 (8.4)	344 (8.0)	2216 (16.3)
White, non-Hispanic	5745	728 (54.2)	1357 (65.3)	3660 (44.5)	23,852	2830 (75.3)	5872 (81.7)	15,150 (65.3)
Black, non-Hispanic	1398	225 (23.4)	183 (11.1)	990 (17.4)	2323	299 (11.2)	320 (5.5)	1704 (9.9)
Other	945	92 (8.8)	118 (8.3)	735 (12.4)	1711	167 (5.2)	261 (4.8)	1283 (8.6)
**Educational Attainment**								
Less than high school	1020	186 (16.2)	183 (10.7)	651 (11.1)	2413	478 (14.1)	523 (7.8)	1412 (9.0)
High school	2390	438 (38.6)	488 (25.1)	1464 (22.5)	8193	1397 (40.4)	1997 (28.7)	4799 (23.1)
Some college	1657	219 (18.5)	390 (21.3)	1048 (17.0)	5407	712 (20.4)	1371 (20.3)	3324 (17.4)
Associate degree	925	114 (10.0)	203 (10.4)	608 (9.2)	3451	398 (10.7)	816 (11.7)	2237 (10.3)
College degree	3670	194 (16.8)	586 (32.6)	2890 (40.2)	11,170	499 (14.4)	2090 (31.6)	8581 (40.2)
**Housing Tenure**								
Public	905	188 (15.5)	194 (9.5)	523 (7.5)	198	44 (1.2)	42 (0.6)	112 (0.6)
Privately owned	2659	203 (17.6)	638 (33.2)	1818 (25.0)	25,684	2522 (69.2)	5926 (85.5)	17,236 (82.0)
Privately rented *	6098	760 (66.9)	1018 (57.4)	4320 (67.5)	4752	918 (29.6)	829 (13.9)	3005 (17.4)

Notes: 1. All percentages are weighted. The complete table with additional variables can be found in Appendix A. 2. Multi-unit housing (MUH) is defined as two or more housing units that share a wall or ceiling. Single-unit housing (SUH) refers to independent housing units, such as single-family homes, mobile homes, and other housing units that do not share a wall or ceiling. 3. *p*-values for all chi-squared tests comparing each characteristic based on housing type and smoking status were statistically significant at *p* < 0.0001. 4. Smoking status was missing for 53 observations with MUH and 153 observations with SUH. Metropolitan status was missing for 60 observations with MUH and 394 with SUH. * Rent includes no cash rent or rent without payment.

**Table 2 ijerph-20-06356-t002:** Association of smoking status with presence of smokefree rules (a) and vapefree rules (b) inside residents’ homes based on housing type, 2019.

	a aOR Presence of Smokefree Rules—CIGARETTES (Complete vs. Partial/None) among Multi- and Single-Unit Housing Residents *			b aOR Presence of Vapefree Rules—E-CIGARETTES (Complete vs. Partial/None) among Multi- and Single-Unit Housing Residents *		
	Unweighted N = 39,624; Weighted N = 218,409,352 aOR (95% CI)			Unweighted N = 39,444; Weighted N = 217,452,139 aOR (95% CI)		
Variable (Reference)	MUH Complete Rule	MUH Partial/No Rule	SUH Complete Rule	MUH Complete Rule	MUH Partial/No Rule	SUH Complete Rule
**Sex (Female)**						
Male	0.95 (0.76–1.20)	1.08 (0.76–1.54)	0.94 (0.77–1.15)	0.91 (0.73–1.14)	1.11 (0.77–1.59)	0.91 (0.76–1.09)
**Age Group (55+)**						
18–24	1.03 (0.59–1.81)	1.30 (0.56–3.01)	1.09 (0.62–1.92)	**0.44 (0.27–0.74)**	1.45 (0.70–3.01)	**0.50 (0.31–0.79)**
25–34	1.16 (0.80–1.67)	0.84 (0.46–1.52)	1.20 (0.85–1.69)	**0.56 (0.39–0.80)**	0.90 (0.52–1.58)	**0.59 (0.43–0.81)**
35–44	1.09 (0.74–1.60)	0.63 (0.34–1.15)	1.37 (0.95–1.98)	0.73 (0.49–1.08)	0.88 (0.52–1.49)	0.94 (0.66–1.36)
45–54	0.87 (0.61–1.24)	0.80 (0.47–1.36)	1.04 (0.77–1.40)	**0.68 (0.48–0.96)**	0.76 (0.43–1.35)	0.80 (0.60–1.07)
**Race/Ethnicity (White, non-Hisp)**						
Hispanic	**2.46 (1.40–4.32)**	1.67 (0.79–3.53)	**1.74 (1.03–2.93)**	**2.63 (1.53–4.53)**	1.54 (0.73–3.25)	**1.83 (1.12–3.00)**
Black, non-Hispanic	0.99 (0.62–1.60)	1.24 (0.69–2.23)	0.74 (0.49–1.10)	1.52 (0.91–2.54)	1.37 (0.74–2.55)	1.14 (0.74–1.76)
Other, non-Hispanic	1.27(0.77–2.10)	1.13 (0.53–2.42)	0.87 (0.53–1.43)	**1.76 (1.02–3.02)**	1.47 (0.71–3.05)	1.22 (0.72–2.06)
**Metropolitan Status (Non-metropolitan)**						
Metropolitan	**3.41 (2.24–5.17)**	**3.25 (1.64–6.41)**	1.14 (0.85–1.53)	**3.29 (2.11–5.15)**	**3.24 (1.67–6.26)**	1.11 (0.81–1.52)
**Education (College degree)**						
Less than high school	**0.25 (0.15–0.41)**	**0.28 (0.14–0.54)**	**0.51 (0.35–0.74)**	**0.36 (0.22–0.61)**	**0.23 (0.12–0.48)**	0.73 (0.48–1.11)
High school	**0.33 (0.23–0.50)**	**0.35 (0.20–0.63)**	**0.59 (0.43–0.82)**	**0.41 (0.28–0.61)**	**0.37 (0.22–0.62)**	0.73 (0.53–1.01)
Some college, no degree	**0.41 (0.27–0.93)**	**0.49 (0.26–0.93)**	**0.71 (0.51–0.99)**	**0.46 (0.30–0.85)**	**0.50 (0.30–0.85)**	0.78 (0.57–1.08)
Associate degree	**0.55 (0.35–0.87)**	0.68 (0.36–1.31)	0.78 (0.53–1.15)	**0.57 (0.37–0.88)**	0.69 (0.37–1.27)	0.81 (0.56–1.19)
**Housing Tenure (Privately owned)**						
Public	**63.25 (18.48–216.48)**	**55.45 (13.62–225.85)**	1.66 (0.51–5.40)	**60.17 (16.50–219.40)**	**58.88 (14.30–242.43)**	1.56 (0.43–5.58)
Privately rented	**8.35 (5.83–11.94)**	**8.33 (4.82–14.42)**	0.74 (0.53–1.03)	**8.64 (6.30–11.85)**	**9.92 (5.67–17.37)**	0.78 (0.58–1.05)
**Smoking Status (Never)**						
Current	**0.08 (0.06–0.10)**	0.83 (0.53–1.30)	**0.09 (0.07–0.12)**	**0.08 (0.06–0.11)**	0.73 (0.47–1.13)	**0.10 (0.08–0.13)**
Former	**0.67 (0.50–0.91)**	1.01 (0.59–1.72)	**0.70 (0.53–0.91)**	**0.37 (0.28–0.49)**	1.09 (0.70–1.70)	**0.40 (0.31–0.51)**

* Results for each multinomial regression model are reported separately in panels Table 2a,b; comparisons are housing/rule categories to SUH Partial/No (smokefree or vapefree) rule, as the referent group. Notes: 1. Models additionally controlled for U.S. Census region, metropolitan status, annual household income, employment status, and occupational category. The complete table can be found in Appendix A. 2. Multi-unit housing (MUH) is defined as two or more housing units that share a wall or ceiling. Single-unit housing (SUH) refers to independent housing units such as single-family homes, mobile homes, and other housing units that do not share a wall or ceiling. Complete rule indicates that respondents reported that no one is allowed to smoke/vape anywhere inside their home. Partial/no rule indicates that respondents reported that smoking/vaping is allowed in some places or at some times inside their home or reported that smoking/vaping is permitted anywhere inside their home, respectively.

## Data Availability

The data from the TUS-CPS are publicly available at https://cancercontrol.cancer.gov/brp/tcrb/tus-cps/questionnaires-data (accessed on 2 September 2022). The data from the ASEC (March Supplement) of the CPS are publicly available at https://www.census.gov/data/datasets/2019/demo/cps/cps-asec-2019.html (accessed on 2 September 2022).

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
