# Peer review of "Disparities in Smokefree and Vapefree Home Rules and Smokefree Policy Attitudes Based on Housing Type and Cigarette Smoking Status, United States, 2019"

_ijerph, 2023, doi:10.3390/ijerph20146356_

Round 1

Reviewer 1 Report

The manuscript presents a comprehensive examination of cigarette smoking status, smoke-free and vape-free home rules, and attitudes towards smoke-free policies at the national level by sociodemographic and housing characteristics using a household-based Current Population Survey. The study found that Multiunit Housing (MUH) residents with current smoking status were 92% less likely to have a complete smoking ban, highlighting disparities in smoke-free and vape-free home rules and smoke-free policy attitudes by housing type and cigarette smoking status.

One concern I have is the overwhelming amount of data resulting from the analyses, which creates a complex situation making it hard to understand the overall picture and what the results mean. Therefore, a clear theoretical framework guiding the analyses and discussion would be beneficial. For example, It would be beneficial to clarify why the types of housing, and whether it is rented or privately owned, should make a difference, as MUH and public housing could be a proxy for low SES.

If possible, I suggest exploring whether non and former smokers who allow smoking are living with a household who smokes, as this could affect other non-smoking households and especially children. Furthermore, while the manuscript provides valuable insights into disparities in smoke-free and vape-free home rules and smoke-free policy attitudes, the results may not be novel, and the discussion section mainly reiterates the results. Providing more specific recommendations on evidence-based policies, prevention, health education, and cessation programs would be beneficial. Additionally, exploring novel initiatives such as peer-support smoking cessation groups in MUH and organizing educational campaigns, as well as using technology for disseminating knowledge, could be suggested.

Author Response

Please see the attachment with updated revisions based on this second round of reviewer comments. We have embedded comment bubbles in the "track changes" version of the document so reviewers can specifically see how their comments were integrated into these revisions.

The manuscript presents a comprehensive examination of cigarette smoking status, smoke-free and vape-free home rules, and attitudes towards smoke-free policies at the national level by sociodemographic and housing characteristics using a household-based Current Population Survey. The study found that Multiunit Housing (MUH) residents with current smoking status were 92% less likely to have a complete smoking ban, highlighting disparities in smoke-free and vape-free home rules and smoke-free policy attitudes by housing type and cigarette smoking status.

One concern I have is the overwhelming amount of data resulting from the analyses, which creates a complex situation making it hard to understand the overall picture and what the results mean. Therefore, a clear theoretical framework guiding the analyses and discussion would be beneficial. For example, It would be beneficial to clarify why the types of housing, and whether it is rented or privately owned, should make a difference, as MUH and public housing could be a proxy for low SES.

Reviewer Response:

We thank the reviewer for this comment. Lines 74-78 in the discussion explain that despite the higher unadjusted prevalence of smoking among residents of publicly rented MUH, our adjusted results show that MUH residents with a complete smokefree rule were over 60 times more likely to live in publicly rented housing compared to residents in a privately rented home. These findings reflect the beneficial impact of the Federal HUD rule on secondhand exposure in public MUH, which does not apply to privately owned MUH. In fact, even privately rented MUH residents fared better than privately owned residents with respect to a complete MUH rule. We have added this additional explanation in the discussion, that these findings by housing tenure reflect adjusted results. Additionally, in our adjusted model, low SES does not appear to be a proxy for the results we observe with respect to MUH residents reporting a complete smokefree rule. Our results show that those living in MUH who were in the lowest educational category were least likely to report a complete smokefree rule.

If possible, I suggest exploring whether non and former smokers who allow smoking are living with a household who smokes, as this could affect other non-smoking households and especially children. Furthermore, while the manuscript provides valuable insights into disparities in smoke-free and vape-free home rules and smoke-free policy attitudes, the results may not be novel, and the discussion section mainly reiterates the results. Providing more specific recommendations on evidence-based policies, prevention, health education, and cessation programs would be beneficial. Additionally, exploring novel initiatives such as peer-support smoking cessation groups in MUH and organizing educational campaigns, as well as using technology for disseminating knowledge, could be suggested.

Response:

We appreciate Reviewer 1’s comment on exploring smoking status and the effect on other non-smoking households including children. However, the TUS did not offer data on household smoking. Regarding more specific recommendations on evidence-based cessation strategies, while the scope of this study does not speak to cessation effectiveness in the context of multiunit housing, we cite other cessation-related studies and resources including the Surgeon General’s 2020 Report on Cessation which contain additional information; accordingly, we state that results of this present study may be able to further inform the development, implementation and sustainability of evidence-based interventions as well as educational campaigns. It is our hope that this study may also catalyze future studies and exploration of novel initiatives. 

Reviewer 2 Report

Thank you for the opportunity to review this manuscript, which uses national data to evaluate home smoking and vaping results, and support for indoor smokefree policies, among people living in single- and multiunit public and private housing. The paper represents a contribution to the literature by using recent national data to illustrate the (pre-covid) state of support for clean indoor air and the ongoing risk of secondhand smoke exposure in multiunit housing. It further provides compelling information regarding the likelihood of exposure to e-cigarette vapor in housing units. However, the paper would benefit from attention to the following comments.  

1.     Abstract: “MUH residents who currently smoked…were least likely to support rules allowing smoking inside all MUH apartments and living areas.” Shouldn’t this be most likely?

2.     Methods: please define “metropolitan status,” e.g. MSA? CBSA? Rural-urban continuum?

3.     Methods: “We also fitted two separate weighted polytomous logistic regression models to examine the association between smoking status and presence of home smoking and vaping rules. Response categories for each home rule var-iable were grouped as a complete ban versus a partial/no ban on either smok-ing or vaping inside the home, among residents of MUH and SUH, sepa-rately. Categories for “partial ban” and “none at all” were combined due to our main interest in examining individuals reporting a complete ban.” Where are these results?

4.     Results: why were SUH and MUH residents lumped together when considering indoor smoking bans in MUH? I’d like to see a separate analysis for MUH residents as these are the people for whom this is a relevant policy issue.

5.     Results: the legends for figures 1 and 2 seem to only encompass the charts on top; include the responses for the indoor smoking ban support question as well

6.     Results: why is SUH partial or no rule used as the reference group for the logistic regressions?

7.    Results: p15 – given the way the modeling is described, then there should be two reference groups for the independent findings, e.g., the statement “Among MUH residents, Hispanic adults were 2.46 times significantly more likely 2 (95% CI, 1.40–4.32) to report a complete smokefree home rule (i.e., no smoking allowed 3 inside the home) versus White non-Hispanics.” Should be, “versus White non-Hispanic adults who live in SUH with a partial/no ban.” And the same for the remaining results. If I’m misinterpreting this, then please describe the analysis and what’s shown in Table 2 more clearly.

8.     Discussion: three  additional limitations that are important to note are 1) TUS doesn’t include an indicator for whether the respondent lives with someone who smokes/vapes/uses tobacco (as far as I know); 2) some of the Ns for current smokers split by tenure are quite small, leading to very large CIs in the adjusted models, and 3) there are substantial differences between duplexes and large apartment complexes in terms of exposure and risk, so classifying all housing other than single unit housing in one category may obscure important differences among housing types.

Author Response

Thank you for the opportunity to review this manuscript, which uses national data to evaluate home smoking and vaping results, and support for indoor smokefree policies, among people living in single- and multiunit public and private housing. The paper represents a contribution to the literature by using recent national data to illustrate the (pre-covid) state of support for clean indoor air and the ongoing risk of secondhand smoke exposure in multiunit housing. It further provides compelling information regarding the likelihood of exposure to e-cigarette vapor in housing units. However, the paper would benefit from attention to the following comments.  

  1. Abstract: “MUH residents who currently smoked…were least likely to support rules allowing smoking inside all MUH apartments and living areas.” Shouldn’t this be most likely?

      Response: Please see Table 2 (Supplemental Table 2). Residents who currently smoke were least likely to support a MUH complete rule (OR = 0.08) compared to former (OR = 0.67) or never smokers (reference group).

  1. Methods: please define “metropolitan status,” e.g. MSA? CBSA? Rural-urban continuum?

      Response: we appreciate the suggestion and have added an explanation in lines 143-144 of the Methods section indicating that metropolitan status is based on CBSA codes.

  1. Methods: “We also fitted two separate weighted polytomous logistic regression models to examine the association between smoking status and presence of home smoking and vaping rules. Response categories for each home rule var-iable were grouped as a complete ban versus a partial/no ban on either smok-ing or vaping inside the home, among residents of MUH and SUH, sepa-rately. Categories for “partial ban” and “none at all” were combined due to our main interest in examining individuals reporting a complete ban.” Where are these results?

Response: We have added a statement in Section 3.3 indicating that selected results are reported in Table 2, due to journal page size restrictions, but the complete table with all results is presented in Supplementary Table 2.

  1. Results: why were SUH and MUH residents lumped together when considering indoor smoking bans in MUH? I’d like to see a separate analysis for MUH residents as these are the people for whom this is a relevant policy issue.

      Response: Please refer to Figure 1. Results for MUH and SUH residents are separated out in light blue background and light green background, respectively.

  1. Results: the legends for figures 1 and 2 seem to only encompass the charts on top; include the responses for the indoor smoking ban support question as well

      Response: Figure 2 only includes illustrated results for home vaping rules because this TUS cycle does not include a question on support for indoor vaping bans.

  1. Results: why is SUH partial or no rule used as the reference group for the logistic regressions?

      Response: We selected SUH partial/no rule as the reference group, as the primary goal of this analysis is to understand smoking prevalence and other factors among MUH residents compared to SUH residents. As such, we believe it makes sense from a secondhand smoke perspective to highlight findings for residents in MUH who reported either a complete smoking rule (followed by a partial/no smoking rule and a SUH complete rule, respectively) compared to the SUH partial/no rule reference group we selected.

  1. Results: p15 – given the way the modeling is described, then there should be two reference groups for the independent findings, e.g., the statement “Among MUH residents, Hispanic adults were 2.46 times significantly more likely 2 (95% CI, 1.40–4.32) to report a complete smokefree home rule (i.e., no smoking allowed 3 inside the home) versus White non-Hispanics.” Should be, “versus White non-Hispanic adults who live in SUH with a partial/no ban.” And the same for the remaining results. If I’m misinterpreting this, then please describe the analysis and what’s shown in Table 2 more clearly.

      Response: These results are based on a polytomous multivariable model, meaning that the outcome variable has >2 categories, in this case 4 levels. One of the 4 groups was selected as the reference category (SUH with no/partial rule). Additionally, each categorical predictor included in the model also has a reference category. These comparisons are presented in the Table 2 as columns (4 outcome levels) and rows (predictors). The first paragraph in section 3.3 explains that the outcome variable reference group is “SUH with a partial or no smokefree rule;” we wanted to avoid a more convoluted explanation of all the outcome group comparisons throughout the writeup of the results. In this way, the reader can focus on the interpretation of each predictor within the binomial comparisons of the outcome variable (e.g., MUH with a complete rule vs. a SUH with partial/no rule).

  1. Discussion: three  additional limitations that are important to note are 1) TUS doesn’t include an indicator for whether the respondent lives with someone who smokes/vapes/uses tobacco (as far as I know); 2) some of the Ns for current smokers split by tenure are quite small, leading to very large CIs in the adjusted models, and 3) there are substantial differences between duplexes and large apartment complexes in terms of exposure and risk, so classifying all housing other than single unit housing in one category may obscure important differences among housing types.

Response: Thank you for these suggestions; they have been incorporated into the draft (please see lines 146-156).

Round 2

Reviewer 1 Report

The paper has shown some improvement but my previous concerns and comments have not been addressed. It remains challenging to read and draw meaningful conclusion from the vast amount of analyses in the absence  of a theoretical framework. 

Author Response

Please see the attachment which contains revisions based on the reviewer comments. We have embedded comment bubbles in the document to specifically point to where our revisions responded to reviewer comments.
